# Interpretable Complex-Valued Neural Networks for Privacy Protection

**Liyao Xiang**[a]**, Hao Zhang**[a]**, Haotian Ma**[b]**, Yifan Zhang**[a]**, Jie Ren**[a]**, and Quanshi Zhang**[a]

[*a]Shanghai Jiao Tong University,      [b]South China University of Technology

## Abstract

Previous studies have found that an adversary attacker can often infer unintended input information from intermediate-layer features. We study the possibility of preventing such adversarial inference, yet without too much accuracy degradation. We propose a generic method to revise the neural network to boost the challenge of inferring input attributes from features, while maintaining highly accurate outputs. In particular, the method transforms real-valued features into complex-valued ones, in which the input is hidden in a randomized phase of the transformed features. The knowledge of the phase acts like a key, with which any party can easily recover the output from the processing result, but without which the party can neither recover the output nor distinguish the original input. Preliminary experiments on various datasets and network structures have shown that our method significantly diminishes the adversary's ability in inferring about the input while largely preserves the resulting accuracy.

## 1 Introduction

Deep neural networks (DNNs) have shown superb capabilities to process massive volume of data, and local devices such as mobile phones, medical equipment, Internet of Things (IoT) devices have become major data entry points in recent years. Although on-device machine learning has exhibited various advantages, it usually burdens thin devices with overwhelming computational overhead. Yet offloading data or processed features to a cloud operator would put the individual privacy at risk. For example, if a user has a virtual assistant at home, it should not worry about its private data being collected and processed by an untrusted cloud operator. The operator, on the other hand, should not be able to recover original signals or their interpretation.

However, as shown in the previous literature (Dosovitskiy & Brox (2016); Zeiler & Fergus (2014); Mahendran & Vedaldi (2015); Shokri et al. (2017); Ganju et al. (2018); Melis et al. (2019)), intermediate-layer features face many privacy threats, where the adversary either reconstructs the input or infers unintended properties about the input. Hence, on-device processing encounters a dilemma, *i.e.* while we expect intermediate-layer features to yield high accuracy, we certainly would not want sensitive information to be leaked.

Therefore, we propose a novel method which tweaks a conventional neural network into a complex-valued one, such that intermediate-layer features are released without sacrificing input privacy too much. More precisely, we turn the original real-valued features into complex-valued ones, rotate these features by a random angle, and feed them to the cloud for further processing.

We face two significant challenges in the design. First, the tweaked features have to be correctly handled by the DNN on the cloud. Although network structures vary depending on the input types, it is desired that the piecewise linearity of features to be preserved, so that the resulting accuracy does not degrade too much compared with the original features. Second, an adversary attacker who intercepts the features in the middle should not be able to recover the exact input, nor can it figure out the correct output. Most importantly, the neural network is supposed to be trained using the original data without additional human supervision, and is efficient to conduct inference.

---

[*]Quanshi Zhang is the corresponding author with the John Hopcroft Center and MoE Key Lab of Artificial Intelligence AI Institute, Shanghai Jiao Tong University, China. `zqs1022@sjtu.edu.cn`

To overcome the first challenge, we tailor operations of DNN to make the succeeding computation at the cloud invariant to feature rotations, *i.e.,* the DNN can correctly handle the feature and preserve the rotated angle in the output. For the second issue, we adopt generative adversarial networks (GAN) to generate synthesized features to hide the original ones. It guarantees $k$-*anonymity* such that, when an adversary tries to recover the original input from the complex-valued features, it would get at least $k$ different inputs (Please see Appendix B for detailed discussion). Each of the $k$ reconstructed inputs has equivalent probabilities of being the original input for the attacker. Apart from the GAN part, the training of the neural network adds little additional computation overhead and requires no extra denotations.

Contributions of this study can be summarized as follows. We propose to transform conventional DNNs into complex-valued ones, which hide input information into a randomly chosen phase. Only the party with the phase information can retrieve the correct output. Without it, any party can neither recover the input nor the output. Tested on various datasets and DNNs, our method has been shown to yield only moderate computational overhead and little accuracy degradation, while significantly diminishes the adversary's ability of inferring about the input.

## 2    RELATED WORK

**Complex-valued neural networks.** Using complex-valued features or parameters in neural networks has always been an interesting topic. From the computational perspective, Trabelsi et al. (2018) showed that complex-valued neural networks have a competitive performance with their real-valued counterparts. By augmenting recurrent neural networks with associative memory based on complex-valued vectors, Danihelka et al. (2016) achieved faster learning on memorization tasks. Complex-valued features can contain information in both the phase and the magnitude. An example is that Reichert & Serre (2013) used the phase to indicate properties of spike timing in cortical information processing. Zhang et al. (2019) used the quaternion-valued neural network to learn 3D-rotation-equivariant features for 3D point cloud processing. We take advantage of complex-valued features such that original features can be hidden in an unknown phase. Also, Yang et al. (2019) shared a similar transformation-based approach as ours while it performed transformation on inputs rather than features, and it aimed to improve the adversarial robustness of the model.

**Privacy-preserving deep learning.** Various privacy-preserving mechanisms have been proposed using different definitions of privacy: Osia et al. (2017) applied the Siamese architecture to separate the primary and private information so that the primary information was preserved in the feature. PrivyNet by Li et al. (2017) was proposed to decide the local DNN structure under the privacy constraint based on the peak signal-to-noise ratio or the pixel-wise Euclidean distance. Data nullification and random noise addition were introduced by Wang et al. (2018) to protect private information in the features, which guaranteed differential privacy.

Cryptographic tools have been used to learn from sensitive data. Zhang et al. (2016) adopted homomorphic encryption, in particular, BGV encryption, to encrypt the private data and perform the high-order back-propagation on the encrypted data; Mohassel & Zhang (2017) distributed the private data among two non-colluding servers who performed the secure two-party computation to train models on the joint data. Our scheme achieves $k$-*anonymity* privacy guarantee such that any adversary can at best reconstruct a number of synthetic inputs from the transformed features and thus it cannot distinguish the original input from others.

## 3    THREAT MODEL AND PRIVACY GUARANTEE

In this section, we briefly introduce the threat model and the privacy guarantee that our scheme aims to achieve.

**Threat model.** In our model, a user processes its raw data locally and transmits the neural network features to the cloud for further processing. An adversary would intercept the transferred features and perform either *feature inversion attacks* or *property inference attacks* on these features to either reconstruct inputs, or infer unintended information. Typical examples of the former attacks include feature inversion by Dosovitskiy & Brox (2016), gradient-based visualization by Zeiler & Fergus (2014); Mahendran & Vedaldi (2015), etc. And the latter ones include membership inference attack

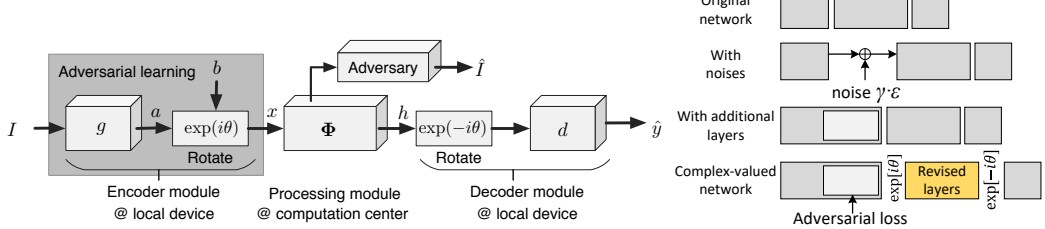

(a) Structure of the complex-valued neural network.   (b) Network structures in experiments.

Figure 1: Design overview and network structures under investigation.

by Shokri et al. (2017), passive property inference attack by Ganju et al. (2018) as well as feature leakage in collaborative learning by Melis et al. (2019).

$k$-**anonymity.** We aim to achieve the following privacy guarantee: when an adversary tries to reconstruct the original input from the transformed features, it can sythesize inputs belonging to $k$ different prototypes with equivalent likelihoods, and only one of them corresponds to the original input.

## 4   APPROACH

### 4.1   OVERVIEW

To fight against the above threats, we propose to revise the conventional DNN. The new structure is shown in Fig. 1a, where the entire network is divided into the following three modules.

- **Encoder** is embedded inside a local device at the user end. The encoder extracts feature from the input, hides true feature by rotating the features by a randomized angle, and sends the encoded result to the cloud.

- **Processing module** is located at the public cloud. This module receives and processes the encoded features without knowing the rotation. At the end of processing, the cloud returns the output to the user.

- **Decoder** resides at the user's local device. The decoder receives and decodes results sent from the cloud to obtain the final result.

The three modules are jointly trained. Once trained, the encoder and decoder are placed at the local device, whereas the processing module resides on the cloud for public use. We will illustrate the detail of each module in the following section.

### 4.2   COMPLEX-VALUED NEURAL NETWORKS

Let $(I, y) \in \mathcal{D}$ denote an input and its label in the training dataset and $g$ be the encoder at the local device. Given the input $I$, the intermediate-layer feature is computed as

$$a = g(I), \tag{1}$$

but we do not directly submit $a$ to the cloud. Instead, we introduce a fooling counterpart $b$ to construct a complex-valued feature as follows:

$$x = \exp(i\theta)\big[a + bi\big], \tag{2}$$

where $\theta$ and $b$ are randomly chosen. $b$ is the fooling counterpart which does not contain any private information of $a$, but its magnitude is comparable with $a$ to cause obfuscation. The encoded feature is then sent to the processing module $\boldsymbol{\Phi}$, which produces the complex-valued feature $h = \boldsymbol{\Phi}(x)$. Upon receiving $h$, the decoder makes prediction $\hat{y}$ on $I$ by inverting the complex-valued feature $h$ back:

$$\hat{y} = d(\Re[h \cdot \exp(-i\theta)]), \tag{3}$$

where $d$ denotes the decoder module, which can be constructed as either a shallow network or just a softmax layer. $\Re(\cdot)$ denotes the operation of picking real parts of complex values.

**Processing Module $\Phi$.** The core design of the processing module is to allow the complex-valued feature $h = \Phi(x)$ to be successfully decoded later by the decoder. *I.e.* if we rotate the complex-valued feature $a + bi$ by an angle $\theta$, all the features of the following layers are supposed to be rotated by the same angle. We represent the processing module as cascaded functions of multiple layers $\Phi(x) = \Phi_n(\Phi_{n-1}(\cdots \Phi_1(x)))$, where $\Phi_j(\cdot)$ denotes the function of the $j$-th layer and $f_j = \Phi_j(f_{j-1})$ represents the output of the $j$-th layer. Thus the processing module should have the following property:

$$\Phi(f^{(\theta)}) = e^{i\theta}\Phi(f) \quad \text{s.t. } f^{(\theta)} \triangleq e^{i\theta}f, \ \forall \theta \in [0, 2\pi). \tag{4}$$

In other words, the function of each intermediate layer in the processing module should satisfy

$$\Phi_j(f_{j-1}^{(\theta)}) = e^{i\theta}\Phi_j(f_{j-1}) \quad \text{s.t. } f_{j-1}^{(\theta)} \triangleq e^{i\theta}f_{j-1}, \ \forall j \in [2, \ldots, n], \ \forall \theta \in [0, 2\pi). \tag{5}$$

to recursively prove Eqn. (4).

Let us consider six most common types of network layers to construct the processing module, *i.e.* the convolutional layer, the ReLU layer, the batch-normalization layer, the average/max pooling layer, the dropout layer, and the skip-connection operation. For the convolutional layer, we remove the bias term and obtain $\text{Conv}(f) = w \otimes f$, which satisfies Eqn. (5). Inspired by Trabelsi et al. (2018), we replace the ReLU with the following non-linear layer:

$$\delta(f_{ijk}) = \frac{\|f_{ijk}\|}{\max\{\|f_{ijk}\|, c\}} \cdot f_{ijk} \tag{6}$$

where $f_{ijk}$ denotes the neural activation at the location $(i, j)$ in the $k$-th channel of the feature, and $c$ is a positive constant.

Likewise, the batch-normalization operation is replaced by

$$\text{norm}(f_{ijk}^l) = \frac{f_{ijk}^l}{\sqrt{\mathbb{E}_l[\|f_{ijk}^l\|^2]}}, \tag{7}$$

where $f^l$ denotes the complex-valued tensor for the $l$-th sample in the batch.

For max-pooling layers, we modify the rule such that the feature with the maximum norm in the region is selected, which ensures that max-pooling does not change the phase of its input.

For the dropout layer, we randomly drops out both the real and imaginary parts of complex-valued features. Skip connections can be formulated as $f + \Phi(f)$, where the inner module $\Phi(f)$ recursively satisfies Eqn. (4).

All above six operations satisfy Eqn. (4). Please see supplementary materials for the proof.

**GAN-based Encoder.** The objective of the encoder is to hide the real feature $a$ of the input $I$ in a certain phase $\theta$ of the encoded feature $x = \exp(i\theta)[a + bi]$. Let $a' = \Re[x\exp(-i\theta')] = \Re[(a + bi)\exp(i\theta - i\theta')] = \Re[(a + bi)\exp(i\Delta\theta)]$ denote a feature decoded using a random angle $\theta' \neq \theta$, where $\Delta\theta = \theta - \theta'$. An ideal encoder requires (i) the decoded feature $a'$ contains sufficient information to cause obfuscation with the real feature $a$; (ii) $a'$ and $a$ follow the same distribution so that it is hard to distinguish which one is the real one. Hence, we train the encoder $g$ with a range of $\Delta\theta$'s and $b$'s, and adopt a GAN to distinguish different values of them. Letting $D$ denote the discriminator, we train the encoder $g$ over the WGAN (Arjovsky et al. (2017)) loss:

$$\min_g \max_D L(g, D) = \mathbb{E}_{I \sim p_\mathcal{I}}\Big[D(a) - \mathbb{E}_{\Delta\theta \sim U(0,\pi), b \neq a}\big[D(a')\big]\Big]$$
$$= \mathbb{E}_{I \sim p_\mathcal{I}}\Big[D(g(I)) - \mathbb{E}_{\Delta\theta \sim U(0,\pi), b \neq g(I)}\big[D\big(\Re[(g(I) + bi)e^{i\Delta\theta}]\big)\big]\Big]. \tag{8}$$

$g$ generates features to fool $D$. Upon convergence, the distribution of $a'$ approximates that of $a$. Note that the loss is the expectation over uniform-randomly distributed $\theta \in (0, 2\pi)$. Ideally, the expectation in Eqn. (8) should also be taken over all possible $\Delta\theta \neq 0$, and thus when an adversary tries to recover $a$ by randomly rotating an angle, it will generate an infinite number of features following the same distribution as that of $a$. Empirically, we pick $k - 1$ such $\Delta\theta$'s uniformly over $(0, \pi)$ and iterate through these $k - 1$ values as negative samples in the adversarial learning. The GAN-based encoder

is the key to our $k$-anonymity guarantee such that it ensures the real feature cannot be distinguished from at least $k - 1$ synthetic features. Please find the end of Appendix B to see the visualization effect of our privacy guarantee.

**The overall loss** of learning is formulated as follows, which contains the adversarial loss in Eqn. (8) and a loss for the target task:

$$\min_{g, \boldsymbol{\Phi}, d} \max_{D} \text{ Loss} = \min_{g, \boldsymbol{\Phi}, d} \big[ \max_{D} L(g, D) + L_{\text{task}}(\hat{y}, y) \big], \tag{9}$$

where $L_{\text{task}}(\hat{y}, y)$ represents the loss for the target task.

Note that to simplify the implementation, we compute $b = g(I')$ as the feature of a randomly chosen sample $I' \neq I$ where ideally $I'$ has little correlation with $I$.

In sum, the entire neural network $[g, \boldsymbol{\Phi}, d]$ is trained with randomized variables $\theta$ and $b$ to minimize the loss in Eqn. (9). Once trained, the network $[g, \boldsymbol{\Phi}, d]$ is fixed and publicly released. At the inference phase, the encoder secretly selects a fooling counterpart $b$ and a rotation angle $\theta$ to encode its real feature $a$. The decoder receives the processed result $h$ from the cloud, and make the final prediction based on Eqn. (3).

## 4.3 ATTACKS TO COMPLEX-VALUED DNNS

We will materialize the attacks described in the threat model in Sec. 3 and make them specific to our proposed complex-valued DNNs.

**Feature inversion attacks:** The adversary performs two types of attacks. In **inversion attack 1**, the adversary tries to find out the most likely rotated angle $\hat{\theta}$ to revert the feature $x$ and obtain $a^*$. An adversary learns to use $a^*$ to reconstruct the input $\hat{I} = \text{dec}(a^*)$. Here we define $a^* = \Re[x \exp(-i\hat{\theta})]$ as the most probable feature recovered by the adversary. The adversary, for instance, can build a new discriminator $D'$ to obtain $\hat{\theta} = \max_{\theta} D'(\Re[x \exp(-i\theta)])$. In **inversion attack 2**, the adversary learns to directly reconstruct the targeted input from the features, *i.e.*, $\hat{I} = \text{dec}(x)$. We show that our adversarial training strategy boosts the difficulty of learning such a decoder. Let $g^*$ and $D^*$ denote the learned encoder and discriminator, respectively. Based on $g^*$, let us construct the following discriminator $\hat{D}(a') = -\|g^*(I) - g^*(\text{dec}(a'))\|$. The larger $\hat{D}(a')$ indicates the higher similarity with the real input. Because $D^*$ is learned via $D^* = \arg\max_D L(g^*, D)$, we have $L(g^*, D^*) \geq L(g^*, \hat{D})$. The adversarial loss in Eqn. (8) provides an upper bound of $L(g^*, \hat{D})$, and thus restricts the capability of $\text{dec}(\cdot)$.

**Property inference attacks:** By Ganju et al. (2018), 'property' refers to the input attribute, which can be any sensitive information to protect or any attribute not expected to be released with features. For example, given a facial image dataset and an age recognition task, the gender attribute is required to hide in features. By labeling features with properties of the corresponding inputs, the adversary is able to train classifiers on a dataset consisting all the feature-property pairs. Once trained, the classifiers can use intermediate-layer features to infer if the input possesses certain properties.

We consider four attacking strategies. **Inference attack 1:** the adversary uses raw images to train a classifier to predict hidden properties of the input. At the testing phase, the attacker feeds the reconstructed result $\text{dec}(a^*)$ from inversion attack 1 to the classifier to predict hidden properties. **Inference attack 2:** the adversary first rotates $x$'s by their respective $\hat{\theta}$ to estimate the most likely feature $a^*$. The adversary trains and tests the classifier on $a^*$ to predict hidden properties. **Inference attack 3:** the adversary uses the result of inversion attack 1 $\text{dec}(a^*)$ to train and test a classifier to predict hidden properties. **Inference attack 4:** different from the aforementioned classification approach, we let the adversary compares $a^*$ against features of each training example to find its $k$-nearest neighbors ($k$-NNs) in the training set and use them to infer the hidden properties.

If $a^*$ contains sufficient information to recover some property, aforementioned attackers would learn the relationship between $a^*$ (or $\text{dec}(a^*)$) and the hidden property. Otherwise, the model would fail.

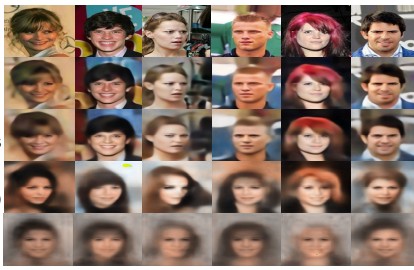
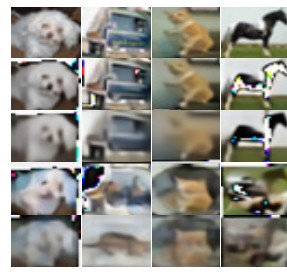

Figure 2: Left: CelebA images reconstructed from different features. From top down: original input, reconstruction from 'encoder' output of the original DNN, $dec(a)$, $dec(a^*)$, $dec(x)$. Right: CIFAR-10 images reconstructed from different features. From top down: original input, reconstruction from 'encoder' output of the noisy DNN, –, $dec(a^*)$, $dec(x)$. Reconstruction has limited resemblance with the input in the last two rows.

| | | Classification Error Rates (%) | | | Reconstruction Errors | | | |
|---|---|---|---|---|---|---|---|---|
| | Dataset | Original DNN | DNN with additional layers | Complex-Valued DNN | Original DNN | DNN with additional layers | Complex-Valued $dec(a^*)$ | Complex-Valued $dec(x)$ |
| ResNet-20-$\alpha$ | CIFAR-10 | 11.56 | 9.68 | 10.91 | 0.0906 | 0.1225 | 0.2664 | 0.2420 |
| ResNet-20-$\beta$ | CIFAR-10 | 11.99 | 9.79 | 12.28 | 0.0967 | 0.1210 | 0.2424 | 0.2420 |
| ResNet-32-$\alpha$ | CIFAR-10 | 11.13 | 9.67 | **10.48** | 0.0930 | 0.1171 | 0.2569 | 0.2412 |
| ResNet-32-$\beta$ | CIFAR-10 | 10.91 | 9.40 | 11.12 | 0.0959 | 0.1189 | 0.2515 | 0.2425 |
| ResNet-44-$\alpha$ | CIFAR-10 | 10.67 | 9.43 | 11.08 | 0.0933 | 0.1109 | 0.2746 | 0.2419 |
| ResNet-44-$\beta$ | CIFAR-10 | 10.50 | 10.15 | **10.51** | 0.0973 | 0.1210 | 0.2511 | 0.2397 |
| ResNet-56-$\alpha$ | CIFAR-10 | 10.17 | 9.16 | 11.53 | 0.0989 | 0.1304 | 0.2804 | 0.2377 |
| ResNet-56-$\beta$ | CIFAR-10 | 10.78 | 9.04 | 11.28 | 0.0907 | 0.1176 | 0.2585 | 0.2358 |
| ResNet-110-$\alpha$ | CIFAR-10 | 10.19 | 9.14 | 11.97 | 0.0896 | 0.1079 | **0.3081** | **0.2495** |
| ResNet-110-$\beta$ | CIFAR-10 | 10.21 | 9.36 | 11.85 | 0.0932 | 0.1152 | 0.2582 | 0.2414 |

Table 1: Classification error rates and reconstruction errors measured on different variants of ResNet and CIFAR-10. Additional layers are introduced due to the GAN structure, but are not trained over adversarial loss. Our results show that the revised DNNs have almost the same utility performance as the original ones, but with greater reconstruction loss.

## 5 EXPERIMENTS

We revised a variety of classical DNNs to complex-valued DNNs and tested them on different datasets to demonstrate the broad applicability of our method. Without loss of generality, tasks include object classification and face attribute estimation, but do not exclude other tasks on DNNs. We have applied a total of two inversion attacks and four inference attacks to the complex-valued DNNs for testing the privacy performance.

### 5.1 IMPLEMENTATION DETAILS

**Complex-valued DNNs:** We constructed complex-valued DNNs based on 8 classical DNNs in total, which included the ResNet-20/32/44/56/110 (He et al. (2016)), the LeNet (LeCun et al. (1998)), the VGG-16 (Simonyan & Zisserman (2015)), and the AlexNet (Krizhevsky et al. (2012)). As shown in Fig. 1a, we divided each original DNN into three modules which were referred to as the encoder/processing module/decoder correspondingly. The structure is given by the top row of Fig. 1b. The bottom row of Fig. 1b gives the architecture of revised DNNs. Given the original DNNs, the encoder/processing/decoder modules were divided as follows. For the residual network, we tested two variants — *ResNet-$\alpha$* and *ResNet-$\beta$* — in the $\alpha$-variant, the output of the layer before the first $16 \times 16$ feature map was fed to the aforementioned GAN, and layers following the first $8 \times 8$ feature map constituted the decoder. The $\beta$-variant was the same with the $\alpha$-variant except that the decoder was composed by the last residual block and the layers following it. The processing module of a residual network was modified such that $c = 1$ in $\delta(\cdot)$ for all non-linear layers.

The encoder of the LeNet consisted of the first convolutional layer and the GAN, whereas its decoder only contained the softmax layer. All layers before the last $56 \times 56$ feature map of VGG-16 comprised the encoder. The decoder consisted of fully-connected layers and the softmax layer. For the AlexNet, the output of the first three convolutional layers was fed into GAN, and the decoder contained fully-connected layers and the softmax layer. In the processing modules of these DNNs, we set $c_k = \mathbb{E}_{ij}\|x'_{ijk}\|$ for neural activations in the $k$-th channel.

| | | Classification Error Rates | | | | | |
| | Dataset | Original DNN | DNN with additional layers | Noisy DNN $\gamma = 0.2$ | Noisy DNN $\gamma = 0.5$ | Noisy DNN $\gamma = 1.0$ | Complex-Valued DNN |
|---|---|---|---|---|---|---|---|
| LeNet | CIFAR-10 | 19.78 | 21.52 | 24.15 | 27.53 | 34.43 | **17.95** |
| LeNet | CIFAR-100 | 51.45 | 49.85 | 56.65 | 67.66 | 78.82 | **49.76** |
| ResNet-56-$\alpha$ | CIFAR-100 | 53.26 | 44.38 | 57.24 | 61.31 | 74.17 | **44.37** |
| ResNet-110-$\alpha$ | CIFAR-100 | 50.64 | 44.93 | 55.19 | 61.12 | 71.31 | 50.94 |
| VGG-16 | CUB-200 | 56.78 | 63.47 | 69.20 | 99.48 | 99.48 | 78.50 |
| AlexNet | CelebA | 14.17 | 9.49 | – | – | – | 15.94 |
| | | Reconstruction Errors | | | | | |
| | Dataset | Original DNN | DNN with additional layers | Noisy DNN $\gamma = 0.2$ | Noisy DNN $\gamma = 0.5$ | Noisy DNN $\gamma = 1.0$ | Complex-Valued DNN $dec(a^*)$ | Complex-Valued DNN $dec(x)$ |
| LeNet | CIFAR-10 | 0.0769 | 0.1208 | 0.0948 | 0.1076 | 0.1274 | **0.2405** | **0.2353** |
| LeNet | CIFAR-100 | 0.0708 | 0.1314 | 0.0950 | 0.1012 | 0.1286 | **0.2700** | **0.2483** |
| ResNet-56-$\alpha$ | CIFAR-100 | 0.0929 | 0.1029 | 0.1461 | 0.1691 | 0.2017 | **0.2593** | **0.2473** |
| ResNet-110-$\alpha$ | CIFAR-100 | 0.1050 | 0.1092 | 0.1483 | 0.1690 | 0.2116 | **0.2602** | **0.2419** |
| VGG-16 | CUB-200 | 0.1285 | 0.1202 | 0.1764 | 0.0972 | 0.1990 | **0.2803** | **0.2100** |
| AlexNet | CelebA | 0.0687 | 0.1068 | – | – | – | **0.3272** | **0.2597** |

Table 2: Classification error rates and reconstruction errors on a variety of DNNs and datasets. We compared the results with noisy DNNs with different noise levels. Complex-Valued DNNs have significantly better accuracy performance and higher reconstruction errors than noisy DNNs over all noise levels.

| | Dataset | Original DNN | w/ additional layers | Noisy DNN $\gamma = 0.2$ | Noisy DNN $\gamma = 0.5$ | Noisy DNN $\gamma = 1.0$ | Complex-Valued DNN $dec(a^*)$ | Complex-Valued DNN $dec(x)$ |
|---|---|---|---|---|---|---|---|---|
| LeNet | CIFAR-10 | 0.16 | 0.12 | 0.20 | 0.20 | 0.24 | **0.82** | **0.92** |
| LeNet | CIFAR-100 | 0.16 | 0.14 | 0.20 | 0.64 | 0.72 | **0.80** | **0.92** |
| ResNet-56-$\alpha$ | CIFAR-100 | 0.06 | 0.06 | 0.08 | 0.10 | 0.36 | **0.72** | **0.88** |
| ResNet-110-$\alpha$ | CIFAR-100 | 0.04 | 0.12 | 0.10 | 0.16 | 0.36 | **0.80** | **0.86** |
| VGG-16 | CUB-200 | 0.06 | 0.06 | 0.08 | 0.02 | 0.14 | **0.86** | **0.84** |
| AlexNet | CelebA | 0.04 | 0.24 | – | – | – | **0.96** | **1.00** |

Table 3: Failure rate of reconstructed images identification. We have five human annotators to recognize the object in mixed groups of images. The results show that most fail to recognize the reconstruction from the revised DNNs.

**Baselines:** Beside the complex-valued DNN, we constructed three baseline networks for comparison.

*Original DNNs (baseline 1):* The original DNN without any revision was taken as the first baseline network. For ease of presentation, we also divided the original DNNs into the encoder, the processing module, and the decoder. The division of modules for original DNNs was the same as that for complex-valued DNNs.

*Noisy DNNs (baseline 2):* Based on the original DNNs, we injected noise right after the encoder module, which was given as $a + \gamma \cdot \epsilon$ (we set $\gamma = 0.5, 1.0, 2.0$). The noisy feature was fed to the processing module instead of $a$, as shown in the second row of Fig. 1b. Here $\epsilon$ represents a high-dimensional random noise with the same average magnitude as $a$.

*DNNs with additional layers (baseline 3):* Since the encoder was based on GAN, additional layers needed to be added, when we revised the DNN. We chose a GAN which incorporated a generator consisting of a convolutional layer with $3 \times 3 \times K$ filters, and a discriminator composed by a convolutional layer as well as a fully-connected layer. We designed this baseline for fair comparison, because additional layers in GAN, although without adversarial loss, would incur model structure changes. Its structure is illustrated by the third row of Fig. 1b.

**Attacks:** Privacy attacking was implemented as follows. (1) *Inversion attacks:* we implemented the inversion attacker based on U-net (Ronneberger et al. (2015)). The original U-net consisted of 8 blocks, and we revised the U-net to construct the attacker. The input feature of the U-net was first up-sampled to the size of the input, which will be fed into the inversion model. There were four down-sampling blocks and four up-sample blocks. Each block contained six convolutional layers for better reconstruction performance. Each down-sample block reduced the size of the feature by half. Each up-sample block doubled the size of the feature map. Thus, the output of the inversion model was of the same size as the input image. (2) *Inference attacks:* we selected CelebA and CIFAR-100 datasets to test robustness against property inference attacks. For CelebA, we evaluated the accuracy performance on classifying 30 attributes and the privacy performance on the rest 10 attributes. Likewise, the accuracy of the DNN on CIFAR-100 was evaluated by the classification error of the major 20 superclasses, and the privacy was gauged by the classification error of the 100 minor classes.

CelebA and CIFAR-100 adopted the AlexNet and the ResNet-56, respectively, as prototype models. Their attacker nets were implemented as the ResNet-50 and the ResNet-56, respectively.

|  | Dataset | Average Error |
|---|---|---|
| ResNet-20-$\alpha$ | CIFAR-10 | 0.7890 |
| ResNet-20-$\beta$ | CIFAR-10 | 0.7859 |
| ResNet-32-$\alpha$ | CIFAR-10 | 0.7820 |
| ResNet-32-$\beta$ | CIFAR-10 | 0.7843 |
| ResNet-44-$\alpha$ | CIFAR-10 | 0.8411 |
| ResNet-44-$\beta$ | CIFAR-10 | 0.7853 |
| ResNet-56-$\alpha$ | CIFAR-10 | 0.8088 |
| ResNet-56-$\beta$ | CIFAR-10 | 0.8283 |
| ResNet-110-$\alpha$ | CIFAR-10 | 0.8048 |
| ResNet-110-$\beta$ | CIFAR-10 | 0.7818 |
| LeNet | CIFAR-10 | 0.7884 |
| LeNet | CIFAR-100 | 0.8046 |
| ResNet-56-$\alpha$ | CIFAR-100 | 0.7898 |
| ResNet-110-$\alpha$ | CIFAR-100 | 0.7878 |
| VGG-16 | CUB-200 | **1.5572** |
| AlexNet | CelebA | 0.8500 |

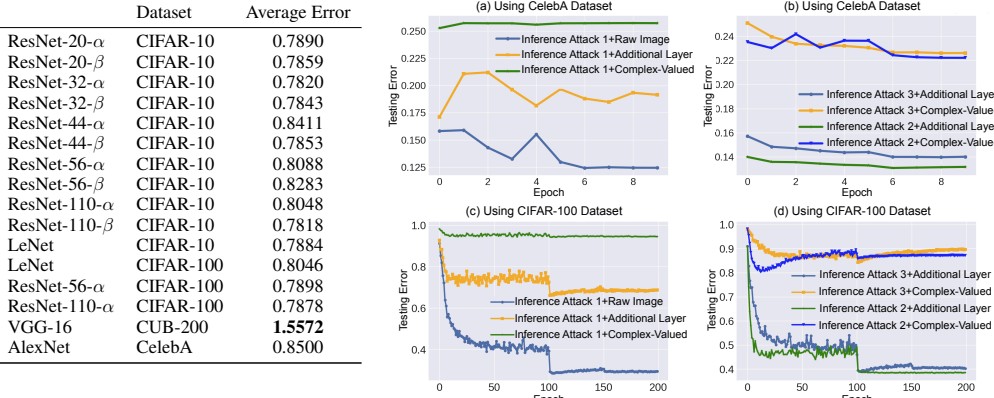

Table 4: Average error (absolute value) of the estimated rotation angle $\theta$ in radian.

Figure 3: The error rates in inferring hidden properties on CelebA (a)/(b) and CIFAR-100 (c)/(d).

| Dataset | DNN structure | Classification Error Rates | Hidden Properties Error Rates | | | Speed(s/images) |
|---|---|---|---|---|---|---|
|  |  |  | 1-NN | 3-NN | 5-NN |  |
| CelebA | w/ additional layers | 0.0804 | 0.2014 | 0.1726 | 0.162 | 0.0026 |
| CelebA | Complex-Valued DNN | 0.1475 | **0.3169** | 0.2790 | 0.2641 | 0.0027 |
| CIFAR-100 | w/ additional layers | 0.1873 | 0.7338 | 0.6837 | 0.7116 | 0.0004 |
| CIFAR-100 | Complex-Valued DNN | 0.2677 | **0.9444** | 0.9363 | 0.925 | 0.0007 |

Table 5: The classification error rates, error rates in inferring hidden properties by $k$-NN, and processing speed. We found that while introducing little computational overhead, our method diminishes the adversary's power in inferring hidden properties.

## 5.2 EVALUATION METRICS

We evaluated three aspects of the revised DNNs' performance: accuracy, privacy, and efficiency. *Accuracy* concerned the performance of the model on the task. *Privacy* was measured by the robustness of our DNNs against inversion attacks and inference attacks. More specifically, for inversion attacks, we had the following measures: (1) the average error of the estimation about the rotated angle $\theta^*$, *i.e.* $|\theta^* - \hat{\theta}|$; (2) the pixel-level reconstruction error $\mathbb{E}[\|\hat{I} - I\|]$ where the value of each pixel was scaled to $[0, 1]$; (3) failure rate of human identification of the reconstructed images. For inference attacks, we measured the robustness of DNNs by the error rate of each attack model (see Sec. 4.3). As to *efficiency*, we measured the processing time compared with the baseline.

## 5.3 EXPERIMENTAL RESULTS AND ANALYSIS

Fig. 2 visualizes images reconstructed using features of different baseline networks. Due to space constraint, this figure only shows partial results. For a complete result, please refer to the supplementary file. Visual effects were significant as the reconstruction in the bottom two rows greatly shifted away from the original input. Obviously, other misleading input attributes were mixed up with the original ones. To further examine the effect, we compared classification error rates and reconstruction errors on a variety of datasets and DNNs in Table 1 and 2. Since the revised DNNs introduced additional layers to the original DNNs, we compared the performance of both the original DNNs as well as those with additional layers. Our complex-valued DNNs achieved similar utility performance, and exhibited significantly higher reconstruction error in almost all groups.

Further, we mimiced the adversary to recover the rotated angle $\theta^*$ with the help of a discriminator model. Most errors fell into $(\pi/4, \pi/2)$ according to Table 4, which indicates even the discriminator could not tell the original features. We also qualitatively tested the reconstructed result in Table 3.

Fig. 3 and Table 5 show the error rates in inferring hidden properties by attackers. We observed that in Fig. 3-(a),(c), the error rate of the inference attack 1 on the complex-valued DNNs ($\text{dec}(a^*)$) was significantly higher than on the baseline 1 (raw images) and baseline 3 ($\text{dec}(a)$), and was close to random selection (99%) on CIFAR-100. We further tested inference attack 2 and 3. In Fig. 3-(b),(d), the error rates of the attacker nets trained on $a^*$ and $\text{dec}(a^*)$ were both significantly higher than that

of baseline 3, *i.e.* $a^*$ indicated little about the hidden properties. $k$-NN methods yielded similar results. Last but not least, we compared the running time of processing the images with different DNNs. Batch size of 128 and 100 are used respectively for CelebA and CIFAR-100. It is shown that the complex-valued DNNs had comparable performance with the baseline.

## 6 CONCLUSION AND DISCUSSIONS

We propose a novel method to preserve input privacy in the intermediate-layer features of deep neural networks. The method transforms a traditional DNN into a complex-valued one. Our method has been tested on a variety of datasets and models. The experimental results have shown the method effectively boosts the difficulty of inferring inputs for the adversary, while largely preserving accuracy for the user. Please also find in the appendices for supplemental experiments.

Theoretically, not only the encoder output but also all other intermediate-layer features in the processing module can preserve privacy. However, we only need to invert the encoder output to test the privacy performance, since all features in successive layers can be expressed as cascaded functions of the encoder output. Thus we may consider the privacy performance of the encoder output as the worst case for all features in the processing module.

## ACKNOWLEDGEMENTS

This work was partially supported by National Natural Science Foundation of China (U19B2043, 61906120, and 61902245), and the Science and Technology Innovation Program of Shanghai (Grant 19YF1424500).

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

# A   PROOF OF $\Phi(e^{i\theta}x) = e^{i\theta}\Phi(x)$

In this paper, we design the processing module to ensure features of all layers are rotated by the same angle, in order to enable the later decryption of the feature.

$$\left. \begin{array}{l} x^{(0)} = a + bi \\ x^{(\theta)} = e^{i\theta}x^{(0)} \end{array} \right\} \Rightarrow \forall j, \; f_j^{(\theta)} = e^{i\theta}f_j^{(0)}$$

where $f_j^{(\theta)}$ and $f_j^{(0)}$ represent the feature map computed using the input $x^{(\theta)}$ and that computed using the input $x^{(0)}$, respectively.

In order to prove the above equation, we revise basic layers/operations in the processing module to ensure

$$\Phi(e^{i\theta}x) = e^{i\theta}\Phi(x)$$

where $\Phi(\cdot)$ denotes the function of a certain layer/operation. $x$ is given as the input of the specific layer/operation $\Phi(\cdot)$. Based on this equation, we can recursively prove $f_j^{(\theta)} = e^{i\theta}f_j^{(0)}$.

Let us consider the following six most common types of layers/functions to construct the processing module, *i.e.* the conv-layer, the ReLU layer, the batch-normalization layer, the average/max pooling layer, the dropout layer, and the skip-connection operation.

1. We revise the conv-layer by omitting the bias term. Thus, we get

$$\Phi(e^{i\theta}x) = w \otimes [e^{i\theta}x] = e^{i\theta}[w \otimes x] = e^{i\theta}\Phi(x)$$

2. We replace the ReLU layer with the non-linear activation function of $\phi(x_{ijk}) = \frac{\|x_{ijk}\|}{\max\{\|x_{ijk}\|, c\}} \cdot x_{ijk}$. Thus, we can write the element-wise operation as follows.

$$\phi(e^{i\theta}x_{ijk}) = \frac{\|e^{i\theta}x_{ijk}\|}{\max\{\|e^{i\theta}x_{ijk}\|, c\}} \cdot [e^{i\theta}x_{ijk}] = e^{i\theta}\left[\frac{\|x_{ijk}\|}{\max\{\|x_{ijk}\|, c\}} \cdot x_{ijk}\right] = e^{i\theta}\phi(x_{ijk})$$

3. We replace the batch-normalization layer with the function of $\phi(x_{ijk}^l) = \frac{x_{ijk}^l}{\sqrt{\mathbb{E}_{ijl}[\|x_{ijk}^l\|^2]}}$. Thus, we can write the element-wise operation as follows.

$$\phi(e^{i\theta}x_{ijk}^l) = \frac{e^{i\theta}x_{ijk}^l}{\sqrt{\mathbb{E}_{ijl}[\|e^{i\theta}x_{ijk}^l\|^2]}} = e^{i\theta}\left[\frac{x_{ijk}^l}{\sqrt{\mathbb{E}_{ijl}[\|x_{ijk}^l\|^2]}}\right] = e^{i\theta}\phi(x_{ijk}^l)$$

4. For the the average/max pooling layer and the dropout layer, we can represent their functions in the form of $\Phi(x) = Ax$, where $x$ is given as a vectorized feature, and $A$ denotes a matrix. For the average/max-pooling operation, $A$ represents the selection of neural activations. For the dropout layer, $A$ contains binary values that indicate the dropout state. In this way, we get

$$\Phi(e^{i\theta}x) = A(e^{i\theta}x) = e^{i\theta}[Ax] = e^{i\theta}\Phi(x)$$

5. For the skip-connection operation, we get

$$\Phi(e^{i\theta}x) = e^{i\theta}x + \Psi(e^{i\theta}x) = e^{i\theta}[x + \Psi(x)] = e^{i\theta}\Phi(x)$$

where $\Psi(\cdot)$ denotes the function that is skipped by the connection. We can recursively ensure $\Psi(e^{i\theta}x) = e^{i\theta}\Psi(x)$.

# B   ARCHITECTURE OF THE GAN

The architecture of GAN is based on the residual network. The generator consists of an input convolutional layer, a residual block from the residual network, and a convolutional layer for the output. The residual block contains two convolutional layers, and a skip connection. Thus, there are four convolutional layers in the generator. The output of the generator is of the same size as the input. The architecture of the discriminator is similar to that of the generator, but the output of the discriminator is a scalar instead of a tensor.

## C  EXPERIMENTS: VISUALIZATION

Fig. 4, Fig. 5 and Fig. 6 respectively give the visualization effect on CIFAR-10, CUB200-2011, and CelebA, when reconstructing the inputs from features.

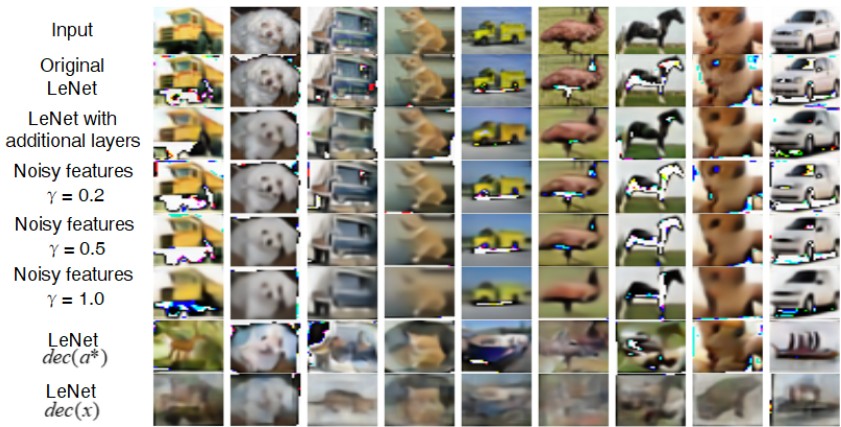

Figure 4: CIFAR-10 images reconstructed from different features on a variety of LeNet-based neural networks.

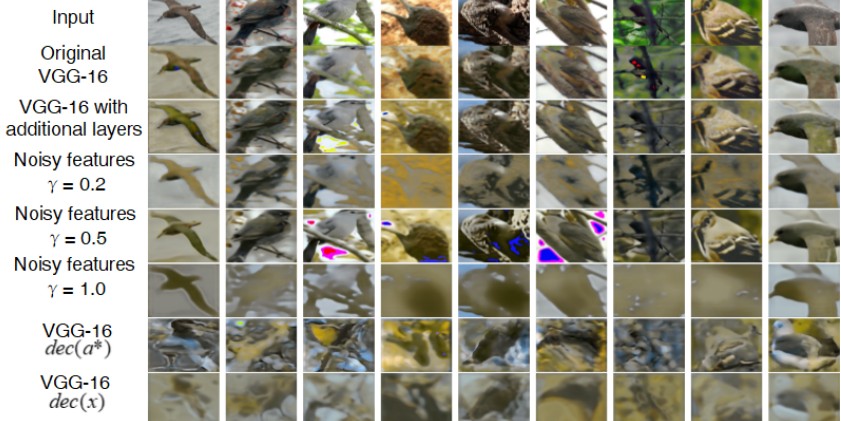

Figure 5: CUB200-2011 images reconstructed from different features on a variety of VGG-16-based neural networks.

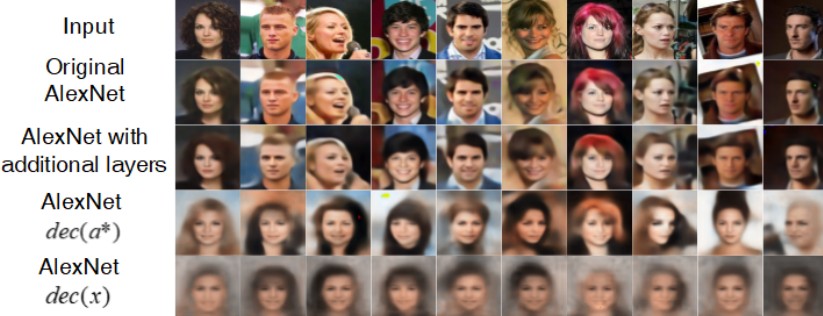

Figure 6: CelebA images reconstructed from different features on a variety of AlexNet-based neural networks.

Fig. 7 shows the reconstructed input images when an adversary launches a feature inversion attack: it rotates $x$ by an angle continuously drawn from $(0, 2\pi)$ and then reconstruct the input on the rotated features. The first image of every two row is the original input image. We found that most meaningful reconstructed pictures are mostly drawn from $(\theta + \frac{\pi}{4}, \theta - \frac{3\pi}{4})$. Most reconstructed images have little

resemblance to the original input, and present meaningful interpolation of the original sample and the randomly chosen sample. The result is an illustration of our $k$-anonymity privacy guarantee.

Figure 7: CelebA images reconstructed from feature $x$ rotated by different angles within $(0, 2\pi)$. We pick the most meaningful ones. The first image of every two row is the original input image.

Fig. 8 shows images reconstructed from correctly decoded features. The result demonstrates that, with the 'key' — $\theta$, one can successfully recover original inputs from features.

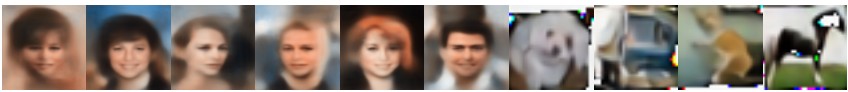

Figure 8: CelebA images and CIFAR-10 images reconstructed from feature $x$ correctly decoded with the knowledge of $\theta$. The reconstructed images have high similarity with the original inputs.

