# OpenReview forum: "Interpretable Complex-Valued Neural Networks for Privacy Protection"
_ICLR.cc/2020/Conference — Accept (Poster)_

### Official Review · AnonReviewer2 · 2019-10-21
**Official Blind Review #2**

**Rating:** 6

**Review:**

After rebuttal,

I really appreciate the authors' effort during the rebuttal, and most of my concerns are addressed well.

===

Summary:

This paper proposed a complex-valued neural network to protect the input data from hidden features of DNNs. Specifically, the authors introduce (1) encoder: producing a complex-valued feature, (2) processing module:  extracting useful features for a decoder and (3) decoder: making a final decision from processed features. Using various deep architectures and datasets, the authors showed that the proposed method can hide the input data from hidden features while maintaining the performance of DNNs.

Detailed comments:

The research topic and main idea of this paper (i.e. introducing a complex-valued neural network for hiding sensitive input data) are interesting and the authors showed that the proposed idea indeed works well using various neural architectures and datasets. It would be more interesting if the authors can consider NLP datasets or other tasks instead of classification. Overall, the paper is well-written and the ideas are novel.

**Experience Assessment:**

I do not know much about this area.

**Review Assessment: Checking Correctness Of Derivations And Theory:**

I carefully checked the derivations and theory.

**Review Assessment: Checking Correctness Of Experiments:**

I carefully checked the experiments.

**Review Assessment: Thoroughness In Paper Reading:**

I read the paper thoroughly.

---

> ### Author Response · Authors · 2019-11-14
> **Response to Reviewer 2**
>
> We thank the reviewer for the positive comments. We would like to answer all the concerns one by one.
>
> Q1: "It would be more interesting if the authors can consider NLP datasets or other tasks instead of classification."
>
> A1: We have followed your suggestions to conduct a new experiment on a neural network for an NLP task.
>
> Since this paper serves as a prototype to verify the effectiveness of complex-valued features in attribute obfuscation, we only use the classification task as an example. We have supplemented a sentiment classification task on Stanford Sentiment Treebank (SST)[1] using the LSTM model [2]. The complex-valued architecture is shown below. The inputs are sentences and outputs are 0/1 labels. The pointwise multiplication operation in LSTM is modified such that the biases are removed: $i_{t} = \sigma(W_{ii}x_{t} + W_{hi}h_{t-1})$.
>
>                                -------------------  a   -----------------------------       -------  d
> X (sentences) -> | Embedding | -> | c=(a + bi)exp(i$\theta$) | -> | LSTM | ->
>                                -------------------       ------------------------------      -------
>  ----------------------      ----------
> | Re(d exp(-i$\theta$))| -> |Classifier| -> Y(0:negative, 1:positive)
>  ----------------------      ----------
> Three sets of experiments are done. First, we train a decoder $dec_1$ by using $a$ to reconstruct the input sentences, and then feed $a$ and $Re(c)$ respectively to $dec_1$ to test the reconstruction accuracy. The results are 87.69% and 68.07% respectively for $a$ and $Re(c)$, verifying that the complex-valued feature indeed preserves input privacy. Second, we train a discriminator $D$ by using $Re(c)$ to find $\hat{\theta} = \arg \max_{\theta} D[Re(c exp(-i\theta))]$. And then we feed $Re(c exp(-i\hat{\theta}))$ to $dec_1$ to reconstruct the input. The testing accuracy is 72.36%. Third, we construct a decoder $dec_2$ by training and testing over $Re(c)$. The testing accuracy is 60.81%.
>
> [1] Richard Socher, Alex Perelygin, Jean Wu, Jason Chuang, Christopher D. Manning, Andrew Ng, and Christopher Potts. 2013. Recursive deep models for semantic compositionality over a sentiment treebank. In EMNLP 2013, pages 1631–1642.
> [2] S. Hochreiter and J. Schmidhuber. Long short-term memory. Neural Computation, 9(8):1735–1780, 1997.

---

### Official Review · AnonReviewer1 · 2019-10-23
**Official Blind Review #1**

**Rating:** 6

**Review:**

Abstract:
In this paper, the authors propose to hide information in phase of the input features. They proposed that if each layer of processing layer, sitting outside of the local unit, is phase preserving, then they can recover the phase back. They propose a modification to the most popular layers in DNN to satisfy that property.

I think the general idea of the paper is interesting, but overall, the paper is very poorly written. It appears that it is written in a rush.

*The abstract and introduction are poorly written. There are poorly written sentences in the text. Here are some examples:
    + "... We propose a generic method to revise a conventional neural network to boost the challenge of adversarially inferring about the input but still yields useful outputs. ...."
    + "... given a transformed feature, an adversary can at best recover a number of features which contain at least another k − 1 features which are different but cannot be distinguished from the real feature.  ..." -- This is so central to the whole paper and it is not well-written. Personally, didn't understand this attack.
Please proofread your paper.

*Why do you need both theta and b ?  It seems to me if \| b \| is comparable to \| a \|, it can destroy the information in original the a arbitrary bad, so it makes sense to keep the norm of b small. However, in the paper, it is suggested to use a random sample (I') and set b = g(I'). I don't see any ablation study in the paper. There is no free lunch, if you provide stronger identity preservation, there should be a compromise in the accuracy, and it seems to be the magnitude of the b is that compromise.

* The whole idea of the GAN encoder is not well justified. What does it mean that the fake feature should contain information "beyond" a ? This very vague.

* "Inference attack 1 " and "Inference attack 4" are the same; only the inference models used in each attack are different. I don't know why the author has separated them.


**Experience Assessment:**

I do not know much about this area.

**Review Assessment: Checking Correctness Of Derivations And Theory:**

I did not assess the derivations or theory.

**Review Assessment: Checking Correctness Of Experiments:**

I assessed the sensibility of the experiments.

**Review Assessment: Thoroughness In Paper Reading:**

I made a quick assessment of this paper.

---

> ### Author Response · Authors · 2019-11-14
> **Response to Reviewer 1**
>
> We greatly appreciate the reviewer's comments. With respect to the writing issues, we have proofread the paper and clarified some statements. We would like to answer all the concerns.
>
> Q1: About English writing. "The abstract and introduction are poorly written ... it is not well-written."
>
> A1: We do understand that the reviewer is mainly bothered by some writing issues. We have revised the statement in the abstract, introduction, and Section 3.
>
> Q2: Concern about technical details of the parameters, theta and b. "Why do you need both theta and b? It seems to me ... the magnitude of the b is that compromise."
>
> A2: In fact, the reviewer's concern may be raised by a misunderstanding about the fooling counterpart b. In the original version of the paper, we have pointed out that the feature decoded by a randomly chosen phase should have the same distribution as the original feature. We also have mentioned that in practice, we could obtain b as the feature of a randomly chosen sample $I' \neq I$ so that the distribution of the synthetic feature $a'$ is close to that of $a$ by minimizing the GAN loss. The norm of $b$ cannot be too small; otherwise, the adversary can recover feature $a$ in spite of the rotation angle. The norm of $b$ should be comparable to the norm of $a$ to cause obfuscation.
>
> Q3: "The whole idea of the GAN encoder is not well justified. What does it mean that the fake feature should contain information beyond a? This very vague."
>
> A3: As mentioned in the paragraph of "GAN-based Encoder" of the original version, the whole purpose of the GAN encoder is to generate synthetic features following the same distribution as that of the original feature, so that an adversary cannot distinguish the original one. To further facilitate understanding, we have revised the paragraph. In particular, we refer to $a'$ as the decoded feature, since it is decoded from x by an adversarially chosen angle.
>
> Q4: "Inference attack 1 and Inference attack 4 are the same; only the inference models used in each attack are different. I don't know why the author has separated them."
>
> A4: This concern is raised due to misinterpretation. Inference attack 1 and 4 are different attacks: attack 1 predicts hidden properties by training a classifier, whereas attack 4 uses  $k$-nearest neighbor method. We would like to verify that despite the specific attack methods, our complex-valued approach prevails in all cases. Nevertheless, we have further clarified details about Inference attack 4 in Section 4.3.

---

### Official Review · AnonReviewer3 · 2019-11-05
**Official Blind Review #3**

**Rating:** 6

**Review:**

This paper proposes a novel way to outsource a part of the information processing in a deep learning model to an untrusted remote location while revealing only little information about the input or final output of the computation. To this end the result of an on-chip encoder (e.g. the first N layers of a ConvNet) is encoded in a complex number with a random phase, which is then shipped to a remote location and gets processed (the next M layers of a ConvNet) in a way that is phase-equivariant. This result is shipped back to the device which extracts the desired information by inverting the phase randomisation. Additional distractor signals are encoded through a GAN-type approach.

Overall the paper is well written although it experiences a few rough edges (e.g. the conclusions, font size in images, typos). The way the problem or threat scenario is phrased in the introduction, however, should emphasise much more the point that none of the actors, be it an adversary intercepting the communication or the cloud operator itself, have to be trusted. In other words, if I have an Alexa device at home which uses this technique I would not have to trust Amazon to keep my processed data private. Instead, even Amazon itself would not be able to really recover the true speech signals or their meaning. In my opinion, this should be a centrepiece of motivation for this work (unless I have overlooked something).

Right now the manuscript focuses on a large set of experiments to empirically demonstrate the effectiveness of the method. What is missing is a more theoretically founded notion of privacy that can yield trustable guarantees. Nonetheless, the novelty of the ideas would still warrant acceptance.

**Experience Assessment:**

I do not know much about this area.

**Review Assessment: Checking Correctness Of Derivations And Theory:**

I assessed the sensibility of the derivations and theory.

**Review Assessment: Checking Correctness Of Experiments:**

I assessed the sensibility of the experiments.

**Review Assessment: Thoroughness In Paper Reading:**

I made a quick assessment of this paper.

---

> ### Author Response · Authors · 2019-11-14
> **Response to Reviewer 3**
>
> We sincerely thank the reviewer for constructive comments, as well as kind suggestions for writing. We have taken the advice and revised the paper for better understanding. We would like to answer all the concerns.
>
> Q1: About English writing. "Overall the paper is well written although it experiences a few rough edges (e.g. the conclusions, font size in images, typos)."
>
> A1: Thank you. We have followed your suggestions and rectified statements. Words in the abstract and the conclusion are rephrased, and fonts in the images are adjusted. Please kindly find the changes in each corresponding section.
>
> Q2: About English writing. "The way the problem or threat scenario is phrased in the introduction, however, should emphasize much more the point that none of the actors, be it an adversary intercepting the communication or the cloud operator itself, have to be trusted. In other words, if I have an Alexa device at home which uses this technique I would not have to trust Amazon to keep my processed data private. Instead, even Amazon itself would not be able to really recover the true speech signals or their meaning. In my opinion, this should be a centerpiece of motivation for this work (unless I have overlooked something)."
>
> A2: In the original version, we have pointed out in the 1st paragraph of the introduction that "Yet offloading raw data to the cloud would put the individual privacy at risk."  For ease of understanding, we have provided a motivating example in the revised introduction.
>
> Q3: "What is missing is a more theoretically founded notion of privacy that can yield trustable guarantees."
>
> A3: The reviewer raises a good point in the theoretical foundation of the privacy guarantee. Despite our efforts, we found that this is due to the lack of a formal definition of privacy in the current literature describing the amount of information that features reveal about the input. Thus, we propose k-anonymity for each release of the features.
>
> We regard each release of the intermediate-layer feature as a release of the corresponding input. According to the definition "a release of data is said to have the k-anonymity property if the information for each entity contained in the release cannot be distinguished from at least k−1 entities of which information also appear in the release," our method guarantees that an adversary cannot distinguish the original input from others by the released feature.
>
> Besides, we have widely surveyed techniques of the feature inversion attack and the property inference attack. We have implemented these attacking techniques in experiments to show the privacy guarantee of our method.

---

### Decision · Program_Chairs · 2019-12-19

**Decision:**

Accept (Poster)

**Comment:**

The reviewers are unanimous in their opinion that this paper offers a novel approach to secure edge learning.  I concur.  Reviewers mention clarity, but I find the latest paper clear enough.